

# Deconvoluting simulated metagenomes: the performance of hard- and soft-clustering algorithms applied to metagenomic chromosome conformation capture (3C)

Matthew Z. DeMaere and Aaron E. Darling

ithree institute, University of Technology Sydney, Sydney, NSW, Australia

## ABSTRACT

**Background**. Chromosome conformation capture, coupled with high throughput DNA sequencing in protocols like Hi-C and 3C-seq, has been proposed as a viable means of generating data to resolve the genomes of microorganisms living in naturally occuring environments. Metagenomic Hi-C and 3C-seq datasets have begun to emerge, but the feasibility of resolving genomes when closely related organisms (strain-level diversity) are present in the sample has not yet been systematically characterised.

**Methods**. We developed a computational simulation pipeline for metagenomic 3C and Hi-C sequencing to evaluate the accuracy of genomic reconstructions at, above, and below an operationally defined species boundary. We simulated datasets and measured accuracy over a wide range of parameters. Five clustering algorithms were evaluated (2 hard, 3 soft) using an adaptation of the extended B-cubed validation measure.

**Results**. When all genomes in a sample are below 95% sequence identity, all of the tested clustering algorithms performed well. When sequence data contains genomes above 95% identity (our operational definition of strain-level diversity), a naive soft-clustering extension of the Louvain method achieves the highest performance.

**Discussion**. Previously, only hard-clustering algorithms have been applied to metagenomic 3C and Hi-C data, yet none of these perform well when strain-level diversity exists in a metagenomic sample. Our simple extension of the Louvain method performed the best in these scenarios, however, accuracy remained well below the levels observed for samples without strain-level diversity. Strain resolution is also highly dependent on the amount of available 3C sequence data, suggesting that depth of sequencing must be carefully considered during experimental design. Finally, there appears to be great scope to improve the accuracy of strain resolution through further algorithm development.

Corresponding author
Aaron E. Darling,
aaron.darling@uts.edu.au

## INTRODUCTION

The explicit and complete determination of the genomes present in an environmental sample is a highly prized goal in microbial community analysis. When combined with their relative abundances, this detailed knowledge affords a great deal of power to downstream investigations in such things as: community metabolism inference, functional ecology, genetic exchange and temporal or inter-community comparison. Unfortunately, the current standard methodology in high-throughput DNA sequencing is incapable of generating data of such exquisite detail, and although raw base-pair yield has increased dramatically with technological progress, a significant methodological source of information loss remains.

The organization of DNA into chromosomes (long-range contiguity) and cells (localization) is almost completely lost as a direct result of two requirements of high-throughput library based sequencing; cell lysis (during the process of DNA purification) and the subsequent shearing (during the sequencing library preparation step). What remains in the form of direct experimental observation is short-range contiguity information. From this beginning, the problem of reestablishing long-range contiguity and reconstructing the original genomic sources is handed over to genome assembly algorithms.

Though the damage done in the steps of purification and fragmentation amounts to a tractable problem in single-genome studies, in metagenomics the whole-sample intermingling of free chromosomes of varying genotypic abundance is an enormous blow to assembly algorithms. Conventional whole-sample metagenome sequencing (*Tringe & Rubin, 2005*) thus results in a severely underdetermined inverse problem (*Venter et al., 2001*; *Myers Jr, 2016*), where the number of unknowns exceeds the number of observations and the degree to which a given metagenome is underdetermined depends variously on community complexity. Accurately and precisely inferring cellular co-locality for this highly fragmented set of sequences, particularly in an unsupervised *de novo* setting, and thereby achieving genotype resolution, remains an unsolved problem.

Recent techniques which repeatedly sample an environment, extracting a signal based on correlated changes in abundance to identify genomic content that is likely to belong to individual strains or populations of cells, have confidently obtained species resolution (*Alneberg et al., 2013*; *Imelfort et al., 2014*) and begun to work toward strain (genotype) resolution (*Cleary et al., 2015*). Inferring abundance per-sample from contig coverage (*Alneberg et al., 2013*; *Imelfort et al., 2014*) or k-mer frequencies (*Cleary et al., 2015*) respectively, the strength of this discriminating signal is a function of community diversity, environmental variation and sampling depth; and represents a significant computational task.

Chromosome conformation capture (3C), a technique first introduced to probe the three-dimensional structure of chromatin (*Dekker et al., 2002*), has become the technological basis for a range of 3C-derived genomic strategies, all of which seek to detect the interaction of spatially proximate genomic loci. The fundamental goal in all cases is to in some way capture a snapshot of the 3D structure of a DNA target.

The methodology begins by fixation (cross-linking) of DNA within intact cells or nuclei, often by formaldehyde, to capture in-place native 3D conformational detail. The nuclei or

cells are lysed and the protein-DNA complexes subjected to restriction digestion to produce free-ends. The resulting complex-bound free-ends are then religated under very low concentration, where conditions favour ligation between free-ends that were in close spatial proximity at the time of fixation. Originally, after this point, signal extraction involved known-primer locus-specific PCR amplification (3C), posing a significant experimental challenge (*De Wit & De Laat, 2012*) and limiting the scale of investigation. To extend its utility, subsequent advances (4C, 5C, HiC) have successively attempted to address the issue of scale by replacing PCR-mediated signal extraction with contemporaneous high-throughput technologies (microarrays, next-generation sequencing (NGS)) (*De Wit & De Laat, 2012*).

The genome-wide strategy of HiC (*Lieberman-Aiden et al., 2009*) exploits NGS to extract interaction signal between all potential sites. To do so, before ligation the method inserts a step in which overhangs are filled with biotinylated nucleotides. Blunt-end ligation is then performed and the DNA purified and sheared. The junction-containing products are then selected for subsequent sequencing by biotin affinity pull-down.

HiC and the closely related meta3C (HiC/3C) have recently been applied to metagenomics (*Beitel et al., 2014*; *Burton et al., 2014*; *Marbouty et al., 2014*), intended as an alternative to purely computational solutions to community deconvolution. Here conventional metagenomic sequencing is augmented with the information derived from HiC/3C read-pairs to provide strong experimental evidence of proximity between genomic loci. This map of interactions greatly increases the power of discrimination between community member genomes, by measuring which sequences were spatially nearby at the time of fixation.

Given sufficient sampling depth, HiC/3C read-pairs have the potential to link points of genomic variation at the genotype level at much longer ranges than has previously been possible (*Selvaraj et al., 2013*; *Beitel et al., 2014*). As with any real experimental process, the generation HiC/3C read-sets is imperfect. Three complications to downstream signal processing are: self-self religations which effectively produce local read-pairs, chimeric read-throughs which span the ligation junction and contain sequence from both ends, and spurious read-pairs involving non-proximity ligation products. Though not insurmountable when integrating HiC/3C data with that of conventional sequencing, these flawed products do at the very least represent a loss of efficiency in generating informative proximity ligation read-pairs.

Sequencing information generated in this way can recover a portion of the information lost in conventional whole genome shotgun (WGS) sequencing. It has been shown that the observational probability of intra-chromosomal read-pairs (*cis*) follows a long-tailed distribution decreasing exponentially with increasing genomic separation (*Beitel et al., 2014*). Inter-chromosomal read-pairs (*trans*), modeled as uniformly distributed across chromosome pairs, typically occur an order of magnitude less frequently than *cis* pairs, and inter-cellular read-pairs are an order of magnitude less frequently again (*Beitel et al., 2014*). This hierarchy in observational probability has the potential to be a precious source of information with which to deconvolute assembled sequence fragments derived from conventionally generated metagenomes into species and perhaps strains.
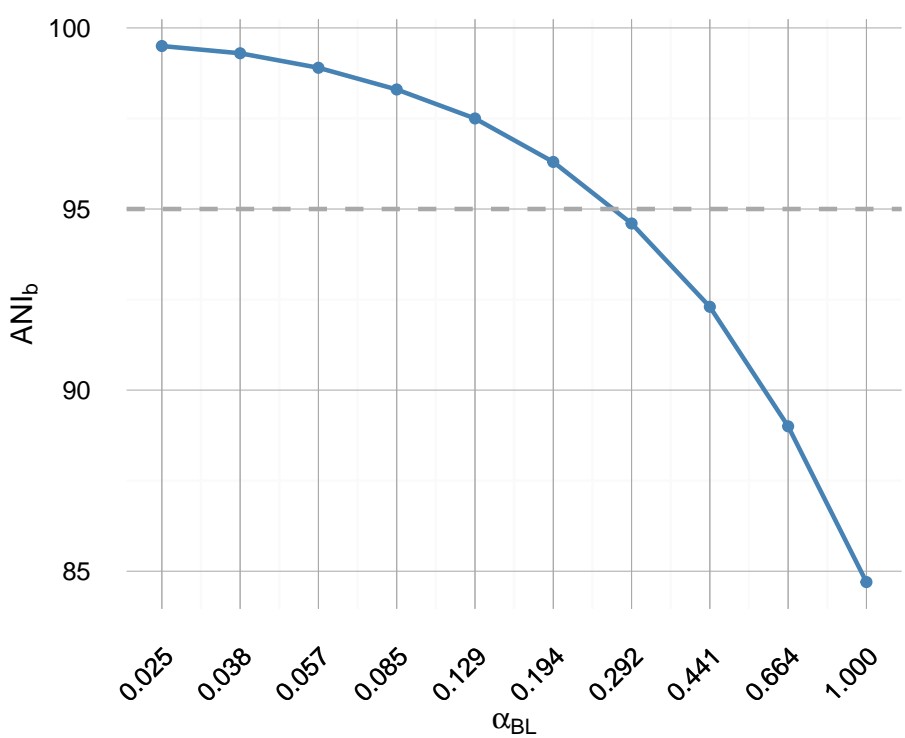

**Figure 1 Average nucleotide identity from BLAST (ANIb) as a function of branch length scale factor $\alpha_{BL}$.** Sampled on a log-scale, the parametric sweep crosses the operational species definition (95% ANIb) roughly midway (dashed grey horizontal line). A 95% similarity threshold is also used internally within IDBA-UD assembler (*Peng et al., 2012*) to determine whether to merge highly similar contigs and has been proposed as a pragmatic definition of bacterial species (*Konstantinidis, Ramette & Tiedje, 2006*; *Richter & Rosselló-Móra, 2009*) akin to 97% 16S rRNA identity.

Previous work which leverages 3C data in assembly analysis has yielded algorithms focused on scaffolding (*Burton et al., 2013*; *Marie-Nelly et al., 2014*). In the context of clonal genome sequencing, 3C directed scaffolding can be applied directly to the entire draft assembly with reasonable success. Beyond monochromosomal genomes, it has been necessary to first cluster (group) assembly contigs into chromosome (plasmid) bins, after which each bin is scaffolded in turn. A move to metagenomics generally entails increased sample complexity and less explicit knowledge about composition. Effectively clustering metagenomic assemblies, containing a potentially unknown degree of both species and strain diversity, represents a challenge that to date has not been thoroughly investigated.

In this work, we describe the accuracy of various analysis algorithms applied to resolving the genomes of strains within metagenomic sequence data. The accuracy of these algorithms was measured over a range of simulated experimental conditions, including varying degrees of evolutionary divergence around our operationally defined species boundary (Fig. 1), and varying depths of generated sequence data. Finally, we discuss implications for the design of metagenomic 3C experiments on systems containing strain-level diversity and describe the limitations of the present work.

## MATERIALS AND METHODS

### Representation

A contact map is formed by mapping proximity-ligation read-pairs to an available reference and counting occurrences between any two genomic regions (*Belton et al., 2012*); where the definition of a genomic region is application dependent. Mathematically, the contact map is a square symmetric matrix $\mathscr{M}$, whose raw elements $m_{ij}$ represent the set of observational frequencies between all genomic regions. The removal of experimental bias by normalization, inference of spatial proximity and finally prediction of chromosome conformation represents the majority of published work in the field to date (*Lieberman-Aiden et al., 2009*; *Noble et al., 2011*; *Yaffe & Tanay, 2011*; *Imakaev et al., 2012*).

Noting that the contact map is equivalent to the weighted adjacency matrix $A$ of an undirected graph $G$ (*Boulos et al., 2013*), an alternative graphical representation is obtained. Here, nodes $n_i$ represent genomic regions and weighted edges $e(n_i, n_j, w_{ij})$ represent the observed frequency $w_{ij}$ of 3C read-pairs linking regions $n_i$ and $n_j$. Expressing the sequencing data as such, a host of graph-theoretic analysis methods can be brought to bear on domain-specific problems.

Possibly the simplest variation, the eponymous 3C-contig graph, defines the genomic regions (and thereby the nodes) to be the set contigs produced by WGS assembly. Fine details such as small indels or single nucleotide variants are not considered with this construction. Even so, the application of the 3C-contig graph to metagenomics (*Beitel et al., 2014*; *Burton et al., 2014*; *Marbouty et al., 2014*) and multichromosomal genome scaffolding (*Burton et al., 2013*) has previously been studied.

The chosen granularity of any construct is a crucial factor in obtaining both sufficiently detailed answers and tractable problems. Though finer scale representations are possible when integrating HiC/3C data into conventional metagenomics, the 3C-contig graph is an effective means of controlling problem scale and can be regarded as a first step toward deeper HiC/3C metagenomic analyses.

### Clustering

Placing entities into groups by some measure of relatedness is often used to reduce a set of objects $O$ into a set of clusters $K$ and ideally where the number of clusters is much less than the number of objects (i.e., $|K| \ll |O|$). When object membership within the set of clusters $K$ is mutually exclusive and discrete, so that an object $o_i$ may only belong to a single cluster $\kappa_k$, it is termed hard-clustering (i.e., $\forall \kappa_k, \kappa_l \in K | k \neq l \rightarrow \kappa_k \cap \kappa_l = \varnothing$). When this condition on membership is relaxed and objects are allowed to belong to multiple clusters, it is termed soft-clustering. The outcome of this potential for multiple membership is cluster overlap, or more formally, that the intersection between clusters $\kappa_k$ and $\kappa_l$ is no longer strictly empty (i.e., $|\kappa_k \cap \kappa_l| \geq 0$).

Possibly motivated by a desire to obtain the plainest answer with maximal contrast, and for the sake of relative mathematical simplicity, hard-clustering is the more widely applied approach. Despite this, many problem domains exist in which cluster overlap reflects real phenomena. For instance, in metagenomes containing closely related species or strains, there is a tendency for the highly conserved core genome to co-assemble in single contigs

while more distinct accessory regions do not. Co-assembly implies that uniquely placing (a 1-to-1 mapping of) contigs into source-genome bins (clusters) is not possible. Rather, an overlapping model is required, allowing co-assembled contigs to be placed multiple times in relation to their degree of source-heterogeneity.

From the aspect of prior knowledge, classification and clustering algorithms fall into three categories (*Jajuga, Sokolowski & Bock, 2002*). Supervised classification, where for a known set of classes, a set of class-labelled objects are used to determine a membership function; semi-supervised classification/clustering, which leverages additional unlabelled data as a means of improving the supervised membership function; and unsupervised clustering, where these prerequisites are not required. Unsupervised algorithms, in removing this *a priori* condition, are preferable if not necessary in situations where prior knowledge is unavailable (perhaps due to cost or accessibility) or the uncertainty in this information is high.

## Appropriate validation measures

Simply put, clustering algorithms attempt to group together objects when they are similar (the same cluster) and separate those objects which differ (different clusters). Although algorithmic complexity can ultimately dictate applicability to a given problem domain, the quality of a clustering solution remains a primary concern in assessing an algorithm's value. To fully assess the quality of a given clustering solution, multiple aspects must be considered. Measures that fail to account for one aspect or another may incorrectly rank solutions. Five important yet often incompletely addressed aspects of clustering quality have been proposed (*Amigó et al., 2009*): homogeneity, completeness, size, number and lastly the notion of a ragbag. Here, a ragbag is when preference is given to placing uncertain assignments in a single catch-all cluster, rather than spreading them across otherwise potentially homogeneous clusters or leaving them as isolated nodes.

External measures, which compare a given solution to a gold-standard are a powerful means of assessing quality and they themselves vary in effectiveness. $F_1$-score, the harmonic mean (Eq. 1) of the traditional measures precision and recall, is frequently employed in the assessment of bioinformatics algorithms. For clustering algorithms, it is perhaps not well known that $F_1$-score fails to properly consider the aspect of completeness (*Amigó et al., 2009*) and further is sensitive to a preprocessing step where clusters and class labels must first be matched (*Hirschberg & Rosenberg, 2007*). The entropy based V-measure (*Hirschberg & Rosenberg, 2007*) was conceived to address these shortcomings but does not consider the ragbag notion nor the possibility of overlapping clusters and classes. The external validation measure Bcubed (*Bagga & Baldwin, 1998*) addresses all five aspects and building from this, extended Bcubed (*Amigó et al., 2009*) supports the notion of overlapping clusters and classes. Analogous to $F_1$-score and V-measure, extended Bcubed is also the harmonic mean of a form of precision and recall.

Still, all of these measures treat the objects involved in clustering as being equal in value when assessing correct and incorrect placements. For some problem domains, it could be argued that correctly classifying object *A* may be more important than correctly classifying object *B*. Conversely, that incorrectly classifying object *A* may represent a larger error than

**Table 1** Clustering algorithm dependent parameters explored in the sweep, where the base set of combinations begins with the fundamental 600 combinations. Only MCL and SR-MCL were swept through additional runtime parameters.

| Algorithm | Name | Description | Type | Number | Total | Values | Sampling |
|-----------|------|-------------|------|--------|-------|--------|----------|
| MCL | infl | Inflation parameter | numeric | 5 | 3,000 | 1.1–2 | linear |
| SR-MCL | infl | Inflation parameter | numeric | 5 | 3,000 | 1.1–2 | linear |
| Louvain-hard | | | | 1 | 600 | | |
| Louvain-soft | | | | 1 | 600 | | |
| OClustR | | | | 1 | 600 | | |

incorrectly classifying object *B*. To this end, we introduce per-object weighting to extended Bcubed (Eq. 1) and propose using contig length (bp) as the measure of inherent value when clustering metagenomic contigs.

## Clustering algorithm selection

Supervised algorithms require *a priori* descriptive detail about the subject of study prior to analysis, while unsupervised algorithms make no such demand. This *a priori* knowledge can be of crucial importance scientifically, such as informing a clustering algorithm how many clusters exist within a dataset under study. For the genome of a single organism, where cluster count corresponds to chromosome count, independent estimation may be tenable. Extracting such descriptive information from an uncultured microbial community in the face of ecological, environmental and historical variation is an onerous requirement. For this reason, we only consider unsupervised algorithms and focus attention on both hard and soft clustering approaches.

Four graph clustering algorithms were considered: MCL, SR-MCL, the Louvain method and OClustR (*Van Dongen, 2001*; *Shih & Parthasarathy, 2012*; *Blondel et al., 2008*; *Pérez-Suárez et al., 2013*). While MCL and Louvain have previously been applied to 3C-contig clustering (*Beitel et al., 2014*; *Marbouty et al., 2014*), to our knowledge SR-MCL and OClustR have not. We did not consider the clustering algorithm employed by (*Burton et al., 2014*) as it requires the number of clusters to be specified *a priori*.

Runtime parameters particular to each algorithm were controlled in the sweep as necessary (Table 1). The widely used MCL (Markov clustering) algorithm (*Van Dongen, 2001*) uses stochastic flow analysis to produce hard-clustering solutions, where cluster granularity is controlled via a single parameter ("inflation"). For this parameter, a range of 1.1 to 2.0 was chosen based on prior work (*Beitel et al., 2014*) and the interval sampled uniformly in five steps (inflation: 1.1–2.0). A soft-clustering extension of MCL, SR-MCL (soft, regularized Markov clustering) (*Shih & Parthasarathy, 2012*) attempts to sample multiple clustering solutions by iterative re-execution of MCL, penalizing node stochastic flows between iterations depending on the previous run state. Beyond MCL's inflation parameter, SR-MCL introduces four additional runtime parameters (balance, quality, redundancy and penalty ratio). It was determined that default settings were apparently

optimal for these additional parameters (results not shown), and therefore only inflation was varied over the same range as MCL.

The Louvain modularity $Q$ (*Newman & Girvan, 2004*) quantifies the degree to which a graph is composed of pockets of more densely interconnected subgraphs. Density is uniform across a graph when $Q = 0$ and there is essentially no community structure, while as $Q \to 1$ it indicates significant community structure with a strong contrast in the degree to which nodes are linked within and between communities. Louvain clustering builds upon this modularity score (*Blondel et al., 2008*), following a greedy heuristic to determine the best partitioning of a graph by the measure of local modularity, identifying sets of nodes more tightly interconnected with each other than with the remainder of the graph. Although a hierarchical solution by recursive application of the Louvain method on the subsequent subgraphs can be obtained, at each step the result is a hard-clustering. We implemented a one-step Louvain clustering algorithm in Python making use of the modules python-louvain and Networkx. We further extended this hard-clustering method (Louvain-hard) to optionally elicit a naive soft-clustering solution (Louvain-soft), where after producing the hard-clustering, any two nodes in different clusters that are connected by an edge in the original graph are made members in both clusters.

We implemented the OClustR algorithm (*Pérez-Suárez et al., 2013*) in Python. The algorithm employs a graph covering strategy applied to a thresholded similarity graph using the notion of node relevance (the average of relative node compactness and density) (*Pérez-Suárez et al., 2013*). The approach functions without the need for runtime parameters, thus avoiding their optimization, and aims to produce clusters of minimum overlap and maximal size.

## Gold standard

A crucial element of external validation is the gold-standard (ground truth). Particularly in the treatment of scientific data, what we call the gold-standard is frequently a "best we can do." Despite the powerful *a priori* advantages gained by the explicit nature of simulation-based studies, practical limitations can introduce uncertainty. In particular, the loss of read placement information in de Bruijn graph assembly means we must infer the genomic origin of each contig rather than obtain it explicitly from assembly output metadata.

In this study, the gold-standard must accurately map the set of assembly contigs $C$ to the set of community source genomes $G$, while supporting the notion of one-to-many associations from contig $c_i$ to some or all genomes $g_i \in G$. It is this one-to-many association that represents the overlap between genomes at low evolutionary divergence. The mapping must also contend with spurious overlap signal from significant local alignments due to such factors as conserved gene content and try to minimize false positive associations.

We used LAST (v712) (*Kiełbasa et al., 2011*) to align the set of assembly contigs $C$ onto the respective set of community reference genomes $G$. For each contig $c_i \in C$, LAST alignments were traversed in order of descending bitscore and used to generate a mask matrix $M$ of contig coverage indexed by both reference genome $g_k \in G$ and contig base position $l$. Rather than a binary representation, mask element $M_{kl}$ was assigned a real value $[0, 1]$

proportional to the identity of the maximal covering alignment to reference genome $g_k$ at site $l$. Lastly, the arithmetic mean $\mu_k$ was calculated over all base positions for each reference genome $g_k$ (i.e., $\mu_k = L_k^{-1}\sum_l M_{kl}$, where $L_k$ is the length of genome $g_k$) and an association between contig $c_i$ and reference genome $g_k$ was accepted if $\mu_k > 0.96$.

## Graph generation

Undirected 3C-contig graphs were generated by mapping simulated 3C read-pairs to WGS assembly contigs using BWA MEM (v0.7.9a-r786) (*Li, 2013*). Read alignments were accepted only in the case of matches with 100% coverage of each read and zero mismatches. In general, this restriction to 100% coverage and identity should be relaxed when working with real data, and we found the iterative strategy employed by (*Burton et al., 2014*) effective in this case (results not shown). Assembly contigs defined the nodes $n_i$ and inter-contig read-pairs the edges ($(n_i, n_j)$ is an edge iff $i \neq j$), while intra-contig read-pairs ($(n_i, n_j) \Leftrightarrow i = j$) were ignored. Raw edge weights $w_{ij}$ were defined as the observed number of read-pairs linking nodes $n_i$ and $n_j$.

## Validation

To assess the quality of clustering solutions a modification to the Extended Bcubed external validation measure (*Amigó et al., 2009*) was made, wherein each clustered object was given an explicit weight. We call the resulting measure "weighted Bcubed" (Eq. 1). For a uniform weight distribution, this modification reduces to conventional Extended Bcubed. In our work, contig length (bp) was chosen as the weight when measuring the accuracy of clustered assembly contigs. Remaining the harmonic mean of Bcubed precision and recall, the weights $w(o_i)$ are introduced here (Eqs. 2 and 3) and the result normalized. For an object $o_i$, the sum is carried out over all members of the set of objects who share at least one class $H(o_i)$ or cluster $D(o_i)$ with object $o_i$ (Eq. 3).

$$F_{b^3} = \frac{2\langle P_{b^3}\rangle \langle R_{b^3}\rangle}{\langle P_{b^3}\rangle + \langle R_{b^3}\rangle} \tag{1}$$

where $\langle P_{b^3}\rangle$ and $\langle R_{b^3}\rangle$ are the weighted arithmetic means of $P_{b^3}(o_i)$ and $R_{b^3}(o_i)$ (Eqs. 2 and 3) over all objects.

$$P_{b^3}(o_i) = \frac{1}{\sum_{o_j \in D(o_i)} w(o_j)} \sum_{o_j \in D(o_i)} w(o_j) P^*(o_i, o_j) \tag{2}$$

$$R_{b^3}(o_i) = \frac{1}{\sum_{o_j \in H(o_i)} w(o_j)} \sum_{o_j \in H(o_i)} w(o_j) R^*(o_i, o_j). \tag{3}$$

Unchanged from Extended Bcubed, the expressions for the Multiplicity Bcubed precision $P^*(o_i, o_j)$ (Eq. 4) and recall $R^*(o_i, o_j)$ (Eq. 5) account for the non-binary relationship between any two items in the set when dealing with overlapping clustering.

$$P^*(o_i, o_j) = \frac{min\left(\left|K(o_i) \cap K(o_j)\right|, \left|\Theta(o_i) \cap \Theta(o_j)\right|\right)}{\left|K(o_i) \cap K(o_j)\right|} \tag{4}$$

$$R^*(o_i, o_j) = \frac{min\left(\left|K(o_i) \cap K(o_j)\right|, \left|\Theta(o_i) \cap \Theta(o_j)\right|\right)}{\left|\Theta(o_i) \cap \Theta(o_j)\right|} \tag{5}$$

where $K(o_i)$ is the set of clusters and $\Theta(o_i)$ is the set of classes for which either contains object $o_i$.

## Simulating HiC/3C read-pairs

A tool for simulating HiC/3C read-pairs was implemented in Python (simForward.py). Read-pairs were generated for a given community directly from its reference genomes, where the relative proportion of read-pairs from a given taxon adhered to the community's abundance profile. Inter-chromosomal (*trans*) pairs were modeled as uniformly distributed across the entire chromosomal extent of a given genome. For intra-chromosomal (*cis*) pairs, a linear combination of the geometric and uniform distributions was used to approximate a long-tailed probability distribution as a function of genomic separation and calibrated by fitting to real experimental data (*Beitel et al., 2014*). For these 3C reads, the modeling of experimental/sequencing error was not performed. Variation in intra-chomosomal probability attributable to 3D chromosomal structure was not included. In effect, chromosomes were treated as flat unfolded rings. The tool takes as input a seed, read length, number of read-pairs, abundance profile and inter-chromosomal probability and outputs reads in either interleaved FastA or FastQ format.

## Pipeline design

The chosen workflow (Fig. 2) represents a simple and previously applied (*Beitel et al., 2014*; *Burton et al., 2014*) means of incorporating 3C read data into traditional metagenomics, via *de novo* WGS assembly and subsequent mapping of 3C read-pairs to assembled contigs. Inputs to this core process are 3C read-pairs and WGS sequencing reads. Outputs are the set of assembled contigs $C$ and the set of "3C read-pairs to contig" mappings $M_{3C}$. Although tool choices vary between researchers, we chose to keep the assembly and mapping algorithms fixed and focus instead on how other parameters influence the quality of metagenomic reconstructions with 3C read data. The A5-miseq pipeline (incorporating IDBA-UD, but skipping error correction and scaffolding via the –metagenome flag) (*Coil, Jospin & Darling, 2015*) was used for assembly. BWA MEM was used for mapping 3C read-pairs to contigs (*Li, 2013*). Parameters placed under control were: WGS coverage (xfold), the number of 3C read-pairs (n3c) and a random seed (S). Prepended to this core process are two preceding modules: community generation and read simulation. The Python implementation of our end-to-end pipeline is available at https://github.com/koadman/proxigenomics.

From a given phylogenetic tree and an ancestral sequence, the community generation module produces a set of descendent taxa with an evolutionary divergence defined by the phylogeny and evolutionary model. The simulated evolutionary process is implemented by sgEvolver (*Darling et al., 2004*), which models both local changes (e.g., single nucleotide substitutions and indels) and larger genomic changes (e.g., gene gain, loss, and rearrangement). The degree of divergence is controlled through a single scale factor $\alpha_{BL}$ (Table S1) that uniformly scales tree branch lengths prior to simulated evolution. As data inputs, the module takes a phylogenetic tree and an ancestral genome. As data outputs, the module generates a set of descendent genomes $G$ and an accompanying gold-standard.

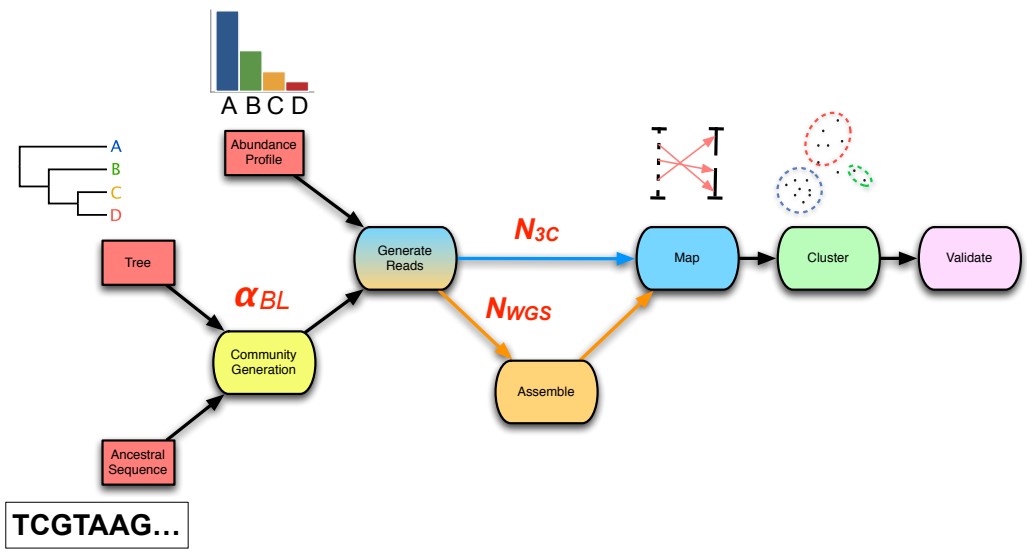

**Figure 2** **The 3C sequencing simulation pipeline used within the parameter sweep.** An ancestral sequence and phylogenetic tree are used in simulating a process of genome evolution with varying divergence ($\alpha_{BL}$). The resulting evolved genomes are subsequently subjected to *in silico* high-throughput sequencing, producing both WGS and 3C read-sets of chosen depth ($N_{WGS}$, $N_{3C}$). WGS reads are assembled and 3C read-pairs are mapped to the resulting contigs to generate a 3C-contig graph. Finally, the graph is supplied to a clustering algorithm and the result validated against the relevant gold-standard.

**Table 2** **Primary parameters under control in the sweep.** In total, each clustering algorithm is presented with 600 combinations which may further increase depending on whether a clustering algorithm also has runtime parameters under control.

| Level | Name | Description | Type | Number | Total | Values |
|---|---|---|---|---|---|---|
| 1 | tree | Phylogenetic tree topology | factor | 2 | 2 | star, ladder |
| 2 | profile | Relative abundance profile | factor | 2 | 4 | uniform, $1/e$ |
| 3 | $\alpha_{BL}$ | Branch length scale factor | numeric | 10 | 40 | 0.025–1 (log scale) |
| 4 | xfold | WGS paired-end depth of coverage | numeric | 3 | 120 | 10, 50, 100 |
| 5 | n3c | Number of 3C read-pairs | numeric | 5 | 600 | 10000, 20000, 50000, 100000, 1000000 |
| 6 | algo | Clustering algorithm | factor | 5 | | MCL, SM-MCL, Louvain-hard, Louvain-soft, OClustR |

Overall, community generation introduces the following two sweep parameters: branch length scale factor $\alpha_{BL}$ and random seed (S) (Table 2).

Following community generation, the read-simulation module takes as input the set of descendent genomes $G$ and generates as output both simulated Illumina WGS paired-end reads and simulated HiC/3C read-pairs. For WGS reads, variation in relative abundance of descendent genomes $G$ was produced by wrapping ART_illumina (v1.5.1) (*Huang et al., 2012*) within a Python script (metaART.py) with the added dependency of an abundance profile table as input. HiC/3C read-pairs were generated from community genomes as outlined above. Generation of the two forms of read-pairs introduces the following sweep parameters: WGS depth of coverage (xfold) and number of 3C read-pairs (n3c) (Table 2).
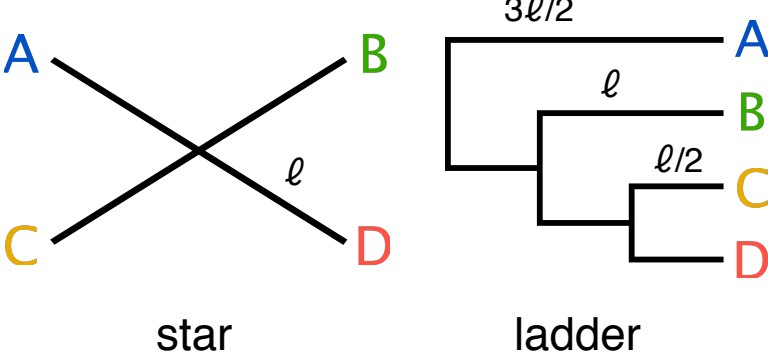

**Figure 3** **Two simple trees of four taxa (A, B, C, D) were used in the parameter sweep.** The star; where all taxa have equal evolutionary distance $\ell$ to their ancestor and ladder; where the distance to the nearest ancestor decreases in incremental steps of $\ell/2$. For the ladder, the length of the internal branch for taxon B was set equal to the branch length of the star and therefore possesses both more closely and more distantly related community members for any value of the scale factor $\alpha_{BL}$ relative to the star topology.

After the assembly and mapping module comes the community deconvolution module, taking as input the set of 3C read mappings $M_{3C}$. Internally, the first step of the module generates the 3C-contig graph $G(n, e, w(e))$. Deconvolution is achieved by application of graph clustering algorithms, where the set of output clusters $K$ reflect predicted genomes of individual community members (*Beitel et al., 2014*; *Burton et al., 2014*).

Lastly, the validation module takes as inputs: a clustering solution, a gold-standard and a set of assembly contigs. The first two inputs are compared by way of weighted Bcubed (Eq. 1), while the set of contigs is supplied to QUAST (v3.1) (*Gurevich et al., 2013*) for the determination of conventional assembly statistics. The results from both clustering and assembly validation are then joined together to form a final output.

## Simulation

Variational studies require careful attention to the number of parameters under control and their sampling granularity, so as to strike a balance between potential value to observational insight and computational effort. Even so, the combinatorial explosion in the total number of variations makes a seemingly small number of parameters and steps quickly exceed available computational resources. Further, an overly ambitious simulation can itself present significant challenges to the interpretation of fundamental system behaviour under the induced changes.

End-to-end, the simulation pipeline makes a large number of variables available for manipulation, and the size and dimensionality of the resulting space is much larger than can be explored with available computational resources. Therefore we decided to focus our initial exploration on a small part of this space. We used two simple phylogenetic tree topologies (a four taxon ladder and a four taxon star) (Fig. 3), to develop insight into the challenges that face metagenomics researchers choosing to apply 3C to communities which contain closely related taxa.

## Parameter sweep

A single monochromosomal ancestral genome was used throughout (*Escherichia coli* K12 substr. MG1655 (acc: NC_000913)). Two simple ultrametric tree topologies of four taxa (tree: star, ladder) (Fig. 3) were included and evolutionary divergence was varied over ten values on a log-scale ($\alpha_{BL}$: 1–0.025; mean taxa ANIb 85–99.5%) (Fig. 1). Two community abundance profiles were tested: uniform abundance and one of decreasing abundance by factors of $1/e$ (i.e., $1, 1/e, 1/e^2, 1/e^3$) (profile: uniform and $1/e$). WGS coverage was limited to three depths (xfold: 10, 50, 100), which for uniform abundance represents 0.12, 0.60 and 1.2 Gbp of sequencing data respectively. Being a simple simulated community, greater depths did not appreciably improve the assembly result. The number of 3C read-pairs (3C sampling depth) was varied in five steps from ten thousand to one million pairs (n3c: 10k, 20k, 50k, 100k, 1M), while the remaining parameter variations can be found in Tables 2 and 1.

From the 40 simulated microbial communities, the resulting 120 simulated metagenome read-sets were assembled and the assemblies evaluated using QUAST (v3.1) (*Gurevich et al., 2013*) against the 20 respective reference genome sets. Both external reference based (e.g., rates of mismatches, Ns, indels) and internal (e.g., N50, L50) statistics were collected and later joined with the results from the downstream cluster validation measures. Data generation resulted in 600 distinct combinations of simulation parameters, forming the basis for input to the selected clustering algorithms. OClustR results in 600 clusterings; Louvain clustering was performed both as standard hard-clustering (Louvain-hard) and our naive soft-clustering modification (Louvain-soft) resulting in 600 clusterings each; lastly MCL and SR-MCL were both varied over one parameter (infl) in five steps resulting in 3000 clusterings each. Finally, the quality of the clustering solutions for all four algorithms was assessed using the weighted Bcubed (Eq. 1) external validation measure. Other parameters fixed throughout the sweep were: ancestral genome size (seq-len: 3 Mbp), indel/inversion/HT rate multiplier (sg_scale: $10^{-4}$), small HT size ($\tilde{P}$oisson(200 bp)), large HT size range ($\tilde{U}$niform(10–60 kbp)), inversion size ($\tilde{G}$eometric(50 kbp)), WGS read generation parameters (read-length: 150 bp, insert size: 450 bp, standard deviation: 100 bp); HiC/3C parameters (read-length: 150 bp, restriction enzyme: NlaIII [ _CATGˆ ]). As simulated genomes were monochromosomal, inter-chromosomal read-pair probability was not a factor.

## Assembly entropy

A normalized entropy based formulation $S_{mixing}$ (Eq. 6) was used to quantify the degree to which a contig within an assembly is a mixture of source genomes, averaged over the assembly with terms weighted in proportion to contig length. For simulated communities, the maximum attainable value is equal to the logarithm of the sum of the relative abundances $q_i$, the effective number of genomes $N_{eff}$ (uniform profile $N_{eff} = 4$, $1/e$ profile $N_{eff} \approx 1.37$). Here $N_C$ is the number of contigs within an assembly, $N_G$ the number of genomes within a community and $L_{asm}$ simply the total extent of an assembly, $p_{ij}$ is the proportion of reads belonging to $i$th genome mapping to the $j$th contig, $l_j$ the length of the $j$th contig, and $h$ the step size in $\alpha_{BL}$.

When each contig in an assembly is derived purely from a single genomic source $S_{mixing} = 0$, conversely when all contigs possess a proportion of reads equal to the relative abundance the respective source genome $S_{mixing} = 1$. A forward finite difference was used to approximate the first order derivative $\Delta S_{mixing}$ (Eq. 7), where mixing was regarded as a function of $\alpha_{BL}$ and the difference taken between successive sample points in the sweep.

$$S_{mixing} = -\frac{1}{L_{asm}\log_2(N_{eff})}\sum_{j=1}^{N_C}l_j\sum_{i=1}^{N_G}p_{ij}\log_2(p_{ij}) \qquad L_{asm} = \sum_{j=1}^{N_C}l_j, \qquad N_{eff} = \sum_{i=1}^{N_G}q_i \qquad (6)$$

$$\Delta S_{mixing}(\alpha_{BL}) = \frac{1}{h}\left(S_{mixing}(\alpha_{BL}+h) - S_{mixing}(\alpha_{BL})\right) \qquad (7)$$

## Graph complexity

Although simple intrinsic graph properties such as order, size and density can provide a sense of complexity, they do not consider the internal structure or information content present in a graph. One information-theoretic formulation with acceptable computational complexity is the non-parametric entropy $H_L$ (Eq. 8) associated with the non-zero eigenvalue spectrum of the normalized Laplacian matrix $N = D^{-1/2}LD^{-1/2}$, where $L = D - A$ is the regular Laplacian matrix, $D$ is the degree matrix and $A$ the adjacency matrix of a graph (*Dehmer & Mowshowitz, 2011*; *Mowshowitz & Dehmer, 2012*; *Dehmer & Sivakumar, 2012*).

$$H_L = \sum_{\lambda_i \in \{\lambda:\lambda>0\}} |\lambda_i|\log_2|\lambda_i| \qquad (8)$$

where $\{\lambda : \lambda > 0\}$ is set the non-zero eigenvalues of the normalized Laplacian $N$.

# RESULTS

## Experimental design

We implemented a computational pipeline which is capable of simulating arbitrary metagenomic HiC/3C sequencing experiments (Fig. 2). The pipeline exposes parameters governing both the process of sequencing and community composition for control by the researcher and further, provides the facility for performing parametric sweeps on these parameters (Table 2).

The pipeline was used to vary community composition, in particular, the degree of within-community evolutionary divergence, and evaluate its impact on the accuracy of genomic reconstruction. Starting from an ancestral sequence, a phylogenetic tree and an abundance profile; 10 communities were generated with varying evolutionary divergence by scaling branch length (Fig. 3). The range of evolutionary divergence was chosen so as to go from a region of easily separable species ($\approx$85% ANI) to that of very closely related strains ($\approx$99.5% ANI) (Fig. 1). The sweep included variation of both WGS coverage (xfold: 10$\times$, 50$\times$, 100$\times$) and the number of HiC/3C read-pairs (n3c: $10^4$–$10^6$) to assess the impact of increased sampling on reconstruction.

Genomic reconstruction was performed using five different graph clustering algorithms (Table 1) on the 600 3C-contig graphs resulting from the sweep. The quality of each solution was then evaluated using our weighted Bcubed metric $F_{B^3}$ (Eq. 1), where the relevant gold-standard as also generated by the pipeline. The resulting dataset is publicly available at http://doi.org/10.4225/59/57b0f832e013c.

## Assembly complexity

Along with traditional assembly validation statistics (N50, L50) (Figs. 4A and 4B), assembly entropy $S_{mixing}$ and its approximate first order derivative $\Delta S_{mixing}$ (Eqs. 6 and 7) (Fig. 4C) were calculated for all 120 combinations resulting from the first four levels of the sweep (parameters: tree, profile, $\alpha_{BL}$, xfold) (Table 2).

As community composition moved from the realm of distinct species ($\alpha_{BL} = 1.0$, ANI$\approx$85%) to well below the conventional definition of strains ($\alpha_{BL} = 0.025$, ANI$\approx$99.5%), the degree of contig mixing increased more or less monotonically, and was delayed by increased read-depth. After $\alpha_{BL}$, the only significant continuous variable influencing mixing was read-depth (Spearman's $\rho = -0.26$, $P < 4 \times 10^{-3}$), while abundance profile was the only significant categorical variable (one-factor ANOVA $R^2 = 0.0774$, $P < 3 \times 10^{-3}$) (Lê, Josse & Husson, 2008). In all cases, as $\alpha_{BL}$ decreased mixing approached unity; implying that as genomic sources became more closely related, the resulting metagenomic assembly contigs were of increasingly mixed origin.

Regarding the assembly process as a dynamic system in terms of evolutionary divergence, the turning point evident in $\Delta S_{mixing}$ (Fig. 4C dashed lines) could be regarded as the critical point in a continuous phase transition from a state of high purity ($S_{mixing} \approx 0$) to a state dominated by completely mixed contigs ($S_{mixing} \rightarrow 1$). This point in evolutionary divergence coincided with the region where assemblies were the most fragmented (max L50, min N50) (Figs. 4A and 4B) and $\Delta S_{mixing}$ was well correlated with both N50 (Spearman's $\rho = 0.72$, $P < 1 \times 10^{-5}$) and L50 (Spearman's $\rho = -0.83$, $P < 1 \times 10^{-7}$), implying that as community divergence decreased through this critical point, traditional notions of assembly quality followed suit.

## Graph complexity

The introduction of 3C sampling depth (number of read-pairs) at the next level within the sweep (parameter: n3c) generated 480 3C-contig graphs (Table 2). To assess how assembly outcome affects the derived graph: order, size, density, and entropy $H_L$ (Eq. 8) were calculated and subsequently joined with the associated factors from assembly (Fig. 4D).

Per the definition of the 3C-contig graph, there was a strong linear correlation between graph order $|n|$ and L50 (Pearson's $r = 0.96$, $P < 3 \times 10^{-16}$) and a weaker but still significant linear correlation between graph size $|e|$ and 3C sampling depth (parameter: n3c) (Pearson's $r = 0.66$, $P < 3 \times 10^{-16}$). Graphical density was moderately linearly correlated with graphical complexity $H_L$ (Pearson's $r = -0.63$, $P < 3 \times 10^{-16}$), and strongly correlated with assembly statistics N50 (Spearman's $\rho = -0.97$, $P < 3 \times 10^{-16}$), L50 (Spearman's $\rho = 0.96$, $P < 3 \times 10^{-16}$) and $\Delta S_{mixing}$ (Spearman's $\rho = -0.73$, $P \leq 1 \times 10^{-16}$).

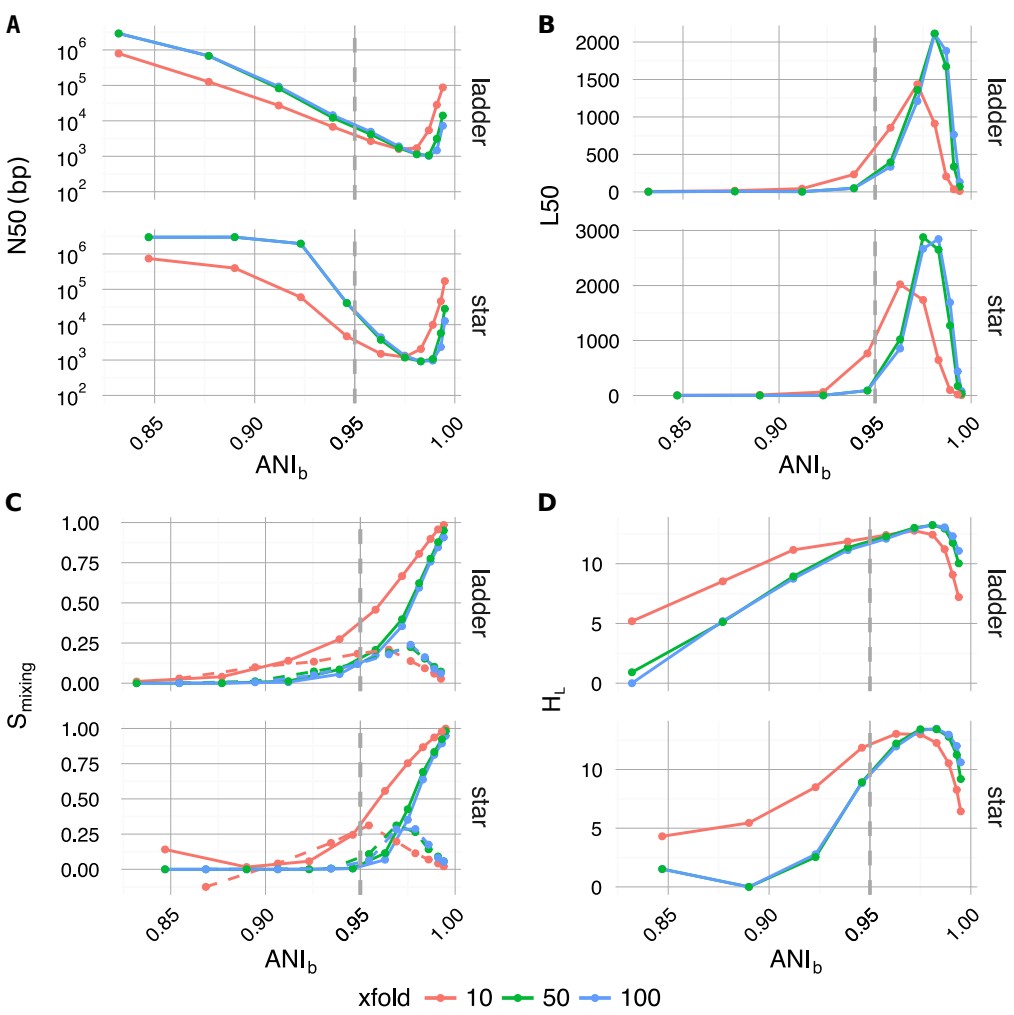

**Figure 4** Plotted as a function of evolutionary divergence (measured by $ANI_b$) for the star and ladder communities at three depths of WGS coverage (10, 50 and 100×); assembly validation statistics N50 (A) and L50 (B), the degree of genome intermixing $S_{mixing}$ and its approximate first order derivative $\Delta S_{mixing}$ (dashed lines) (C), lastly graphical complexity $H_L$ (D). The vertical grey dashed line in each panel marks our operationally defined species boundary ($ANI_b = 95\%$). As evolutionary divergence decreased from easily separable species ($ANI_b \approx 85\%$) to very closely related strains ($ANI_b \to 1$), assemblies went through a transition from a state of high purity ($S_{mixing} \approx 0$) to a highly degenerate state ($S_{mixing} \approx 1$), where many contigs were composed of reads from all community members. A crisis point was observed for small evolutionary divergence ($\alpha_{BL} < 0.2924$, $ANI_b < 95\%$), where a sharp change in contiguity (implied by N50 and L50) occured. At very low divergence, N50 and L50 statistics implied that assemblies were recovering, while source degeneracy ($S_{mixing}$) monotonically increased. Graphical complexity ($H_L$) exhibited a similar turning point to L50 and was dominated by graph order $|n|$ (number of contigs/nodes).

The knock-on effect of evolutionary divergence on the 3C-contig graphs derived from metagenomic assemblies was clear; fragmented assemblies comprised of contigs of mixed heritage resulted in increased 3C-contig graph complexity. As 3C read-pairs are the direct observations used to infer an association between contigs, it could be expected that the correlation between 3C sampling depth and graphical size ($|e|$) would be high ($r \to 1$). In fact, we observed a more moderate correlation ($r = 0.66$) and, because spurious read-pairs

were excluded in our simulations, what might be perceived as a shortfall in efficiency was simply the accumulation of repeated observations of read pairs linking the same contig pairs. Therefore by the nature of the experiment, increased 3C sampling depth did not lead to increased graphical complexity in the same way that a more fragmented assembly would. Instead, increased 3C sampling depth can significantly improve the quality of clustering solutions by increasing the probability of observing rare associations and repeat observations of existing associations.

## Clustering validation

The 300 contig graphs resulting from the sweep at uniform abundance were used to assess the influence of the various parameters on the performance of five clustering algorithms. For each clustering algorithm, overall performance scores, using $F_{b^3}$ (Eq. 1), were joined with their relevant sweep parameters and PCA performed in R (FactoMineR v1.32) (*Lê, Josse & Husson, 2008*). The first three principal components explained 75% of the variation, where PC1 was primarily involved with factors describing graphical complexity ($\alpha_{BL}$: $r = 0.91$, $P < 2 \times 10^{-118}$; density: $r = 0.67$, $P < 8 \times 10^{-41}$; order: $r = -0.75$, $P < 3 \times 10^{-56}$; ANIb: $r = -0.91$, $P < 2 \times 10^{-117}$; $H_L$: $r = -0.91$, $P < 3 \times 10^{-113}$), PC2 factors described the sampling of contig-contig associations and overall connectedness of the 3C-contig graph (size: $r = 0.84$, $P < 2 \times 10^{-79}$; n3c: $r = 0.84$, $P < 7 \times 10^{-82}$; modularity: $r = -0.40$, $P < 9 \times 10^{-13}$) and PC3 pertained to local community structure (modularity: $r = 0.73$, $P < 1 \times 10^{-49}$; and xfold: $r = 0.53$, $P < 3 \times 10^{-23}$) (Fig. 5).

Of the five clustering algorithms, the performance of four (MCL, SR-MCL, Louvain-hard and OClustR) was strongly correlated with PC1 and so their solution quality was inversely governed by the degree of complexity in the input graph, which in turn was largely influenced by within-community evolutionary divergence. The fifth algorithm, our naive Louvain-soft, though also correlated with PC1 and so negatively affected by graphical complexity, possessed significant correlation with PC2 ($r = 0.53$, $P < 5 \times 10^{-23}$) and thus noticeably benefited from increasing the number of 3C read-pairs (Fig. 5).

## DISCUSSION

By selecting a slice from within the sweep and the best-scoring runtime configuration for each algorithm, a qualitative per-algorithm comparison of clustering performance under ideal conditions can be made (Fig. 6). For evolutionary divergence well above the level of strains and prior to the critical region of assembly ($\alpha_{BL} \gg 0.292$, $ANI_b \ll 95\%$), all algorithms achieved their best performance ($F_{b^3} \rightarrow 1$) (Fig. 6C). As evolutionary divergence decreased toward the level of strains and the assembly process approached the critical region, a fall-off in performance was evident for all algorithms and this performance drop is largely attributable to the loss of recall (Fig. 6B). Hard-clustering algorithms (MCL, Louvain-hard) in general exhibited superior precision (Fig. 6A) to that of soft-clustering algorithms (SR-MCL, OClustR, Louvain-soft) and the precision of soft-clustering algorithms was worst in the critical region where graphical complexity was highest.

A hundred-fold increase in the number of 3C read-pairs ($10^4 - 10^6$) had only a modest effect on clustering performance for four of the five algorithms, the exception being

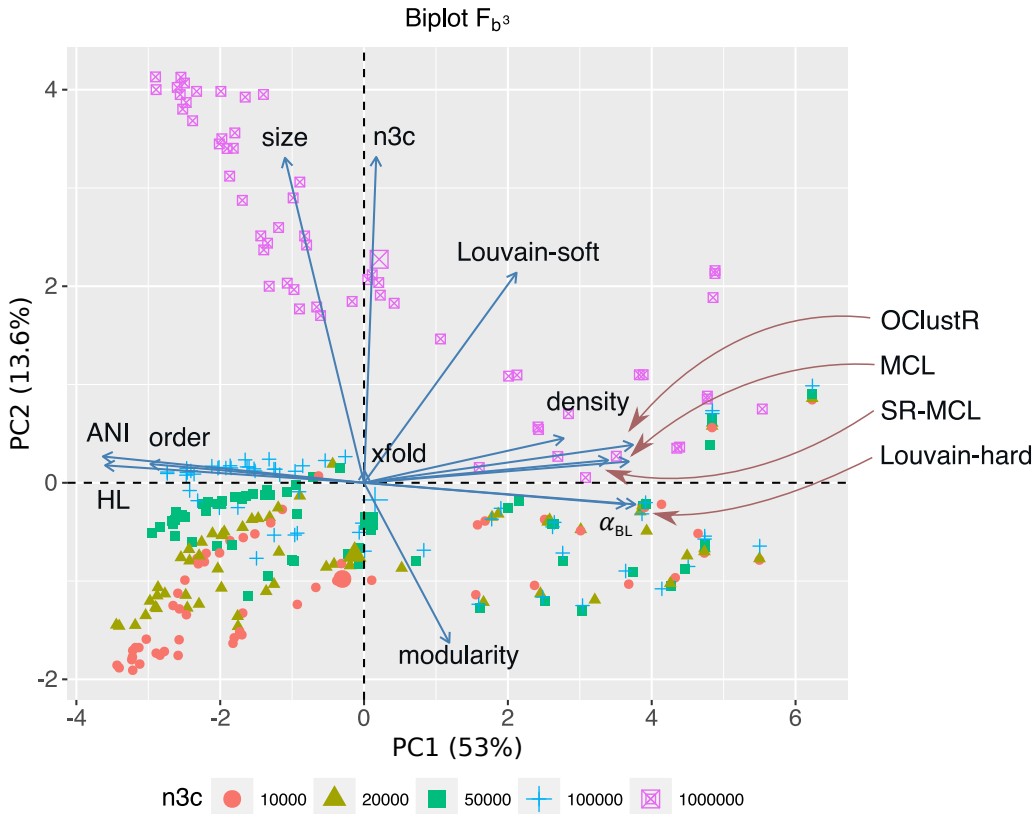

**Figure 5** **For the 300 3C-contig graphs pertaining to uniform abundance, a PCA biplot is shown for the two most significant components (PC1, PC2).** Respectively, PC1 and PC2 explain 53% and 13.6% of the variation within the data-set. Here vectors represent sweep variables, while points represent individual 3C-contigs graphs and are coloured by 3C sequencing depth (n3c: 10k–1M pairs). Double-sized points show mean values of these n3c groupings. Vectors labelled after the five clustering algorithms represent performance as measured by scoring metric $F_{B^3}$ (Eq. 1). PC1 and PC2 explained 53% and 13.6% of the variation within the data-set respectively. PC1 was most strongly correlated with graphical complexity ($H_L$) and the number of graph nodes (*order*), which come about with decreasing evolutionary divergence ($ANI_b$ and $\alpha_{BL}$) and explained the majority of variation in performance for four out of five clustering algorithms. The notable exception was Louvain-soft which had significant support on PC2. PC2 was related to HiC/3C sampling depth (*n3c*), which correlated strongly with the number of graph edges (*size*). The positive response Louvain-soft had to increasing the number of HiC/3C read-pairs (*n3c*) relative to the remaining four algorithms is evident.

our naive Louvain-soft. Louvain-soft made substantial gains in recall from increased HiC/3C sampling depth at evolutionary divergences well below the level of strains ($\alpha_{BL} < 0.085$, $ANI_b < 98\%$), but sacrificed precision at large and intermediate evolutionary divergence. The soft-clustering SR-MCL also sacrificed precision but failed to make similar gains in recall as compared to Louvain-soft. Recall for all three hard-clustering algorithms (MCL, Louvain-hard, OClustR) decreased with decreasing evolutionary divergence as the prevalence of degenerate contigs grew. This drop in recall was particularly abrupt for the star topology where, within the assembly process, all taxa approached the transitional region simultaneously. Being primarily limited by their inability to infer overlap, increase in 3C sampling depth for the hard-clustering algorithms had little effect on recall.

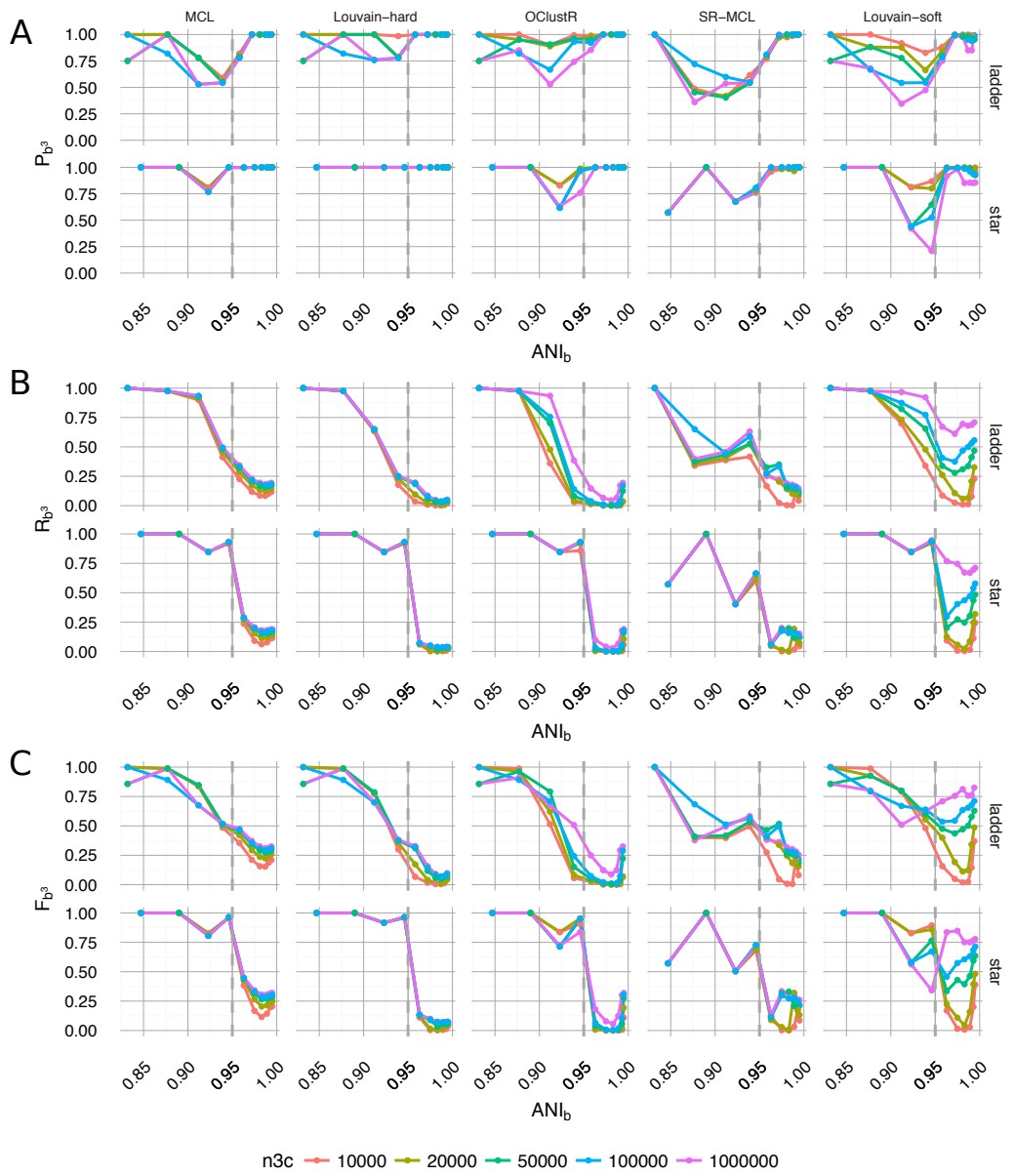

**Figure 6** **Performance of the five clustering algorithms (MCL, Louvain-hard, OClustR, SR-MCL, Louvain-soft), as measured by weighted extended Bcubed precision $P_{b^3}$ (A), recall $R_{b^3}$ (B) and their harmonic mean $F_{b^3}$ (C) (Eqs. 1–3).** The slice from the sweep pertained to uniform abundance and $100 \times$ WGS coverage and the best performing runtime parameters specific to algorithms (i.e., for MCL and SR-MCL inflation = 1.1). (A) Louvain-hard demonstrated high precision throughout, while our simple modification Louvain-soft lead to a drop, particular in the region of intermediate evolutionary divergence. (B) All algorithms struggled to recall the four individual genomes as evolutionary divergence decreased and cluster overlap grew. Within the region of overlap, Louvain-soft performed best and clearly benefited from increasing the number of HiC/3C read-pairs (n3c: $10^4 - 10^6$). (C) In terms of $F_{b^3}$, the harmonic mean of Recall and Precision, only Louvain-soft appeared to be an appropriate choice when it might be expected that strain-level diversity exists within a microbial community.
Our results have implications for the design of metagenomic 3C sequencing experiments. When genomes with >95% ANI exist in the sample, the power to resolve differences among those genomes can benefit greatly from the generation of additional sequence data beyond what would be required to resolve genomes below 95% ANI. In our experiments, the best results were achieved with 100× WGS coverage and 1 million HiC/3C read-pairs. For the simple communities of four genomes each of roughly 3 Mbp considered here, 100× coverage corresponds to generating approximately 1.2 Gbp of Illumina shotgun data. In a metagenomic 3C protocol (*Marbouty et al., 2014*), obtaining 100,000 proximity ligation read-pairs would require approximately $10^7$ read-pairs in total; when we assume a proximity ligation read-pair rate of 1% (*Liu & Darling, 2015*). We note that current Illumina MiSeq V3 kits are specified to produce up to $\approx 2 \times 10^7$ read-pairs, while HiSeq 2500 V4 lanes are specified to yield up to $\approx 5 \times 10^8$ read-pairs per lane. Therefore, while it may be possible to resolve closely related genomes in very simple microbial communities with the capacity of a MiSeq, the scale of the HiSeq is likely to be required in many cases. Alternatively, the more technically complicated HiC protocol (*Beitel et al., 2014*) may be advantageous to achieve higher proximity ligation read rates, with up to 50% of read pairs spanning over 1kbp.

## Limitations and future work

Our simulation of 3C read-pairs did not include modeling of experimental noise in the form basic sequencing error nor spurious ligation products that do not reflect true DNA:DNA interactions. Such aberrant products have been estimated to occur in real experiments at levels up to 10% of total yield in 3C read-pairs (*Liu & Darling, 2015*). As a first approximation, we feel it reasonable to assume that these erroneous read-pairs are a result of uniformly random ligation events between any two DNA strands present in the sample. The sampling of any such spurious read-pair will be sparse in comparison to the spatially constrained true 3C read-pairs and therefore amount to weak background noise. As currently implemented, the Louvain-soft clustering method would be prone to creating false cluster joins in the presence of such noise, but a simple low-frequency threshold removal (e.g., requiring some constant number $N$ links to join communities instead of 1) could in principle resolve the problem.

Only 3C read-pairs were used when inferring the associations between contigs, while conventional WGS read-pairs were used exclusively in assembly. It could be argued that also including WGS read-pairs during edge inference would have had positive benefits, particularly when assemblies were highly fragmented in the critical region. Simulated communities were chosen to be particularly simple for the sake of downstream comprehension. Larger and more complex phylogenetic topologies are called for in fully assessing real-world performance. For the entire sweep, only a single ancestral genome (*Escherichia coli* K12 substr. MG1655) was used in generating the simulated communities and its particular characteristics will have biased genome assembly and sequence alignment tasks within the work-flow. As future work, a more thorough sampling of available microbial genomes and more complicated community structures could be investigated.

Only raw edge weights were used in our analysis because normalization procedures, such as have been previously employed (*Beitel et al., 2014*; *Marbouty et al., 2014*; *Burton et al., 2014*), proved only weakly beneficial at higher 3C sampling depths and occasionally detrimental in situations of low sampling depth (results not shown). For higher sampling depth, the weak response can likely be attributed to a lack of complexity and the low noise environment inherent in the simulation. For low sampling depth, observation counts are biased to small values (mode $\left[w(n_i, n_j)\right] \to 1$) and simple counting statistics would suggest there is high uncertainty ($\pm\sqrt{w(n_i, n_j)}$) in these values. As such, this uncertainty is propagated via any normalization function $f(w(n_i, n_j))$ that attempts to map observation counts to the real numbers ($f : \mathbb{N} \to \mathbb{R}$). Even under conditions for high sampling depth, pruning very infrequently observed low-weight edges can prove beneficial to clustering performance as, beyond this source of uncertainty, some clustering algorithms appear to unduly regard the mere existence of an edge even when its weight is vanishingly small relative to the mean.

For the sake of standardization and to focus efforts on measuring clustering algorithm performance we elected to use a single assembly and mapping algorithm. However, many alternative methods for assembly and mapping exist. In the case of assembly, there are a growing number of tools intended explicitly for metagenomes, such as metaSPAdes (*Bankevich et al., 2012*), MEGAHIT (*Li et al., 2015*), or populations of related genomes (Cortex) (*Iqbal, Turner & McVean, 2013*), while the modular MetAMOS suite (*Treangen et al., 2013*) at once offers tantalising best-practice access to the majority of alternatives. For HiC/3C analysis, a desirable feature of read mapping tools is the capability to report split read alignments (e.g., BWA MEM) (*Li, 2013*). Because of the potential for 3C reads to span the ligation junction, mappers reporting such alignments permit the experimenter the choice to discard or otherwise handle such events. Though we explored the effects of substituting alternative methods to a limited extent (not shown), both in terms of result quality and practical runtime considerations, a thorough investigation remains to be made.

The present implementation state of the simulation pipeline does not meet our desired goal for ease of configuration and broader utility. Of the numerous high-throughput execution environments (SLURM, PBS, SGE, Condor) in use, the pipeline is at present tightly coupled to PBS and SGE. It is our intention to introduce a grid-agnostic layer so that redeployment in varying environments is only a configuration detail. Although a single global seed is used in all random sampling processes, the possibility for irreproducibility remains due to side-effects brought on by variance in a deployment target's operating system and codebase. Additionally, though the pipeline and its ancillary tools are under version control, numerous deployment-specific configuration settings are required post checkout. Preparation of a pre-configured instance within a software container such as Docker would permit the elimination of many such sources of variance and greatly lower the configurational barrier to carrying out or reproducing an experiment.

Many commonly used external validation measures (e.g., F-measure, V-measure) have traditionally not handled cluster overlap and were inappropriate for this study. Ongoing development within the field of soft-clustering (also known as community detection in networks) has, however, led to the reformulation of some measures to support

overlap (*Lancichinetti, Fortunato & Sz, 2009*) or re-expression of soft-clustering solutions into a non-overlapping context (*Xie, Szymanski & Liu, 2011*). While a soft-clustering reformulation of normalized mutual information (NMI) (*Lancichinetti, Fortunato & Sz, 2009*) has become frequently relied on in clustering literature (*Xie, Kelley & Szymanski, 2013*), alongside Bcubed the two have been shown to be complementary measures (*Jurgens & Klapaftis, 2013*). Therefore, although the choice to rely on the single measure we proposed here (Eq. 1) is a possible limitation, it simultaneously avoids doubling the number of results to collate and interpret.

We chose to limit the representation of the combined WGS and 3C read data to a 3C-contig graph. While other representations built around smaller genomic features, such as SNVs, could in principle offer greater power to resolve strains, they bring with them a significant increase in graphical complexity. How more detailed representations might impact downstream algorithmic scaling, or simply increase the difficulty in accurately estimating a gold-standard remains to be investigated.

Benchmark graph generators (so called LFR benchmarks) have been developed that execute in the realm of seconds (*Lancichinetti, Fortunato & Radicchi, 2008*; *Lancichinetti & Fortunato, 2009*). Parameterizing the mesoscopic structure of the resulting graph, their introduction is intended to address the inadequate evaluation of soft-clustering algorithms, which too often relied on unrepresentative generative models or *ad hoc* testing against real networks. Our pipeline may suffice as a pragmatic, albeit much more computationally intensive means of generating a domain specific benchmark on which to test clustering algorithms. Whether it is feasible to calibrate the LFR benchmarks so as to resemble 3C graphs emitted by our pipeline could be explored. Ultimately, the parameter set we defined for the pipeline (Table 2) has the benefit of being domain-specific and therefore meaningful to experimental researchers.

Detection of overlapping communities in networks is a developing field and much recent work has left the performance of many clustering algorithms untested for the purpose of deconvolving microbial communities via 3C read data. Not all algorithms are wholly unsupervised. Individually they fall into various algorithmic classes (i.e., clique percolation, link partitioning, local expansion, fuzzy detection and label propagation). Label propagation methods have shown promise with respect to highly overlapped communities (*Xie, Szymanski & Liu, 2011*; *Chen et al., 2010*; *Gaiteri et al., 2015*), which we might reasonably expect to confront when resolving microbial strains. Empirically determined probability distributions, such as those governing the production of intra-chromosomal (*cis*) read-pairs as a function of genomic separation, might naturally lend themselves to methods from within the fuzzy-detection class. With a generative community model in hand, exploring the performance of gaussian mixture models (GMM), mixed-membership stochastic block models (SBM) or non-negative matrix factorization (NMF) could be pursued.

The incomplete nature of graphs derived from experimental data can result in edge absence or edge weight uncertainty for rare interactions, with the knock-on effect that clustering algorithms can then suffer. We have shown that increasing 3C sampling depth (Fig. 6) can significantly improve the quality of the resulting clustering solutions. A

computational approach, which could potentially alleviate some of the demand for increased depth has been proposed (EdgeBoost) (*Burgess, Adar & Cafarella, 2015*) and shown to improve both Louvain and label propagation methods, is a clear candidate for future assessment.

## CONCLUSION

For a microbial community, as evolutionary divergence decreases within the community, contigs derived from WGS metagenomic assembly increasingly become a mixture of source genomes. When combined with 3C information to form a 3C-contig graph, evolutionary divergence is directly reflected by the degree of community overlap. We tested the performance of both hard and soft clustering algorithms to deconvolute simulated metagenomic assemblies into their constituent genomes from this most simple 3C-augmented representation. Performance was assessed by our proposed weighted variation of extended Bcubed validation measure (Eq. 1), where here weights were set proportional to contig length. We have shown that soft-clustering algorithms can significantly outperform hard-clustering algorithms when intra-community evolutionary divergence approaches a level traditionally regarded as existing between microbial strains. In addition, although increasing sampling depth of 3C read-pairs does little to improve the quality of hard-clustering solutions, it can noticeably improve the quality of soft-clustering solutions. Of the tested algorithms, the precision of the hard-clustering algorithms often equalled or exceeded that of the soft-clustering algorithms across a wide range of evolutionary divergence. However, the poor recall of hard-clustering algorithms at low divergence greatly reduces their value in genomic reconstruction. We recommend that future work focuses on the application of recent advances in soft-clustering methods.

## ACKNOWLEDGEMENTS

We thank Steven P. Djordjevic for his support and helpful discussions. This work was made possible by the AusGEM initiative, a collaboration between the NSW Department of Primary Industries and the ithree institute. We acknowledge the use of computing resources from the NeCTAR Research Cloud, an Australian Government project conducted as part of the Super Science initiative and supported by the EIF and NCRIS.

### Funding

This work was supported under Australian Research Council's Discovery Projects funding scheme (project number LP150100912). The funders had no role in study design, data collection and analysis, decision to publish, or preparation of the manuscript.

### Grant Disclosures

The following grant information was disclosed by the authors:
Australian Research Council's Discovery Projects funding scheme: LP150100912.

## Competing Interests

The authors declare there are no competing interests.

## Author Contributions

- Matthew Z. DeMaere conceived and designed the experiments, performed the experiments, analyzed the data, contributed reagents/materials/analysis tools, wrote the paper, prepared figures and/or tables, reviewed drafts of the paper.
- Aaron E. Darling conceived and designed the experiments, contributed reagents/materials/analysis tools, reviewed drafts of the paper.

## Data Availability

Source code hosted on Github: https://github.com/koadman/proxigenomics.git

Data hosted by our institutions longterm archival service UTS Research Data. DOI 10.4225/59/57b0f832e013c.

## Supplemental Information

Supplemental information for this article can be found online at http://dx.doi.org/10.7717/peerj.2676#supplemental-information.

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
