# Peer review of "Deconvoluting simulated metagenomes: the performance of hard- and soft- clustering algorithms applied to metagenomic chromosome conformation capture (3C)"

_PeerJ, doi:10.7717/peerj.2676_

## Round 0.1 · original submission · Major Revisions

I have received two thorough reviews of your paper and both reviewers agree that your paper is of interest to our readership and offers a solid study on an interesting topic. However, both reviewers identify a number of issues that you will need to deal with before your paper can move forward. Please carefully consider these reviews. I concur with their general assessment and details of suggestions. Good luck in your revision.

Reviewer 1 ·

Basic reporting

In general, the paper is well written. It is self-contained and gives a comprehensive review of the relevant methods and previous work.

The most important Figures 4 and 6 are placed at the very end of the paper, rather than close to relevant paragraph making them slightly harder to follow. Also, Figure 6 is overloaded with data, making it hard to interpret and match to the textual description.

Experimental design

The paper studies the performance of different graph clustering algorithms applied to the problem of binning metagenomic assembly contigs based on 3C data linkage.
I think that the described research falls into the scope of the journal.

As discussed in the paper, most previous works on metagenomic binning with 3C (HiC) data have been using graph clustering approaches. At the same time, performance of those approaches has not been thoroughly assessed in presence of closely-related genomes. This study fills this knowledge gap by assessing performance of different clustering algorithms on extensive set of simulated datasets. Thus I find discussed problem is relevant and meaningful.

Validity of the findings

1. It is not clear why the authors sometimes implement 3rd party algorithms themselves, while there are readily available implementations. For example, on line 208, they mention their implementation of OClustR algorithm (P ́erez-Su ́arez et al., 2013) in Python, while Perez and Suarez provide implementation in C++. Why not just use this implementation?

2. As far as I understood, the raw data has not been made available, but the instructions to reproduce the simulations are given in the paper. Unfortunately, relevant scripts have not been provided to the reviewers.

3. For a fixed set of parameters, only a single dataset has been generated. Having more replicates should increase the robustness of the findings.

In general, the findings confirm reasonable expectations, but show interesting particular boundaries on the  applicability of different popular methods.

Additional comments

Major comments:

1. The question of “sufficient” 3C data depth was left beyond the scope. Would 10^6 HiC paired-reads give even better results for soft-clustering approach? When should the saturation be expected?

2. Related important question: if the perfect linkage data was given, how would clustering algorithms perform? It seems that testing algorithms in a perfect conditions first is crucial for understanding of their limitations!

3. It seems that “ladder” phylogeny is largely useless! Interpretation of corresponding results is much harder and the community is still far too simple to be presented as an approximation to the real case.

4. In general, the impressive complexity of the methods and measures used (definitely a strong side of this work) seems to be in disproportion with the simplicity of the setting which is being analysed. 4-strain mixtures used in the study appear to be a significant oversimplification of strain-mixtures in real metagenomic communities. Would the increase in number of strains change the result?

5. Principal components analysis interpretation is a bit cryptic and looks like an overkill.

6. Moreover, increase in HiC depth also leads to increased graph complexity!

7. Should not weight of the edges in the contig graph be normalized by the contig lengths?

8. The paper refer to clustering algorithms with a given number of clusters as supervised algorithms. Traditionally, supervised learning means that there is some training data set, is that the case for their studies? Although for K-means clustering determining the value of K can be an issue, usually it refers to unsupervised or semi-supervised learning algorithms.


Minor comments and misprints:

1. Introduction states that “Computational approaches to genome assembly would have little success if not for the … assumption that a very high degree of sequence identity (>95% ANI) implies a common chromosomal origin.” The sentence is not clear. Reviewer is unaware of genome assemblers that use this assumption.

2. On line 98, it should be “been” rather than “be”.

3. Beginning of third page states “including varying degrees of evolutionary divergence around the species boundary”, while “species boundary” is not explained up until much later in the text. Reader would benefit from explaining it here.

4. Logical formula on line 128 lacks some quantifiers and uses \subset in place of \in. Logical formula on line 130 is a tautology (uses \supset in place of \ne).

5. The paper states:
"For instance, in the metagenomic analysis of closely related species or strains, the tendency of the highly conserved core genome to co-assemble into single contigs while the more distinct accessory genomes tend not to, implies that a 1-to-1 correspondence of cluster-to-genome is not possible and an overlapping model is required."
Here, it is not clear how contigs are related to clusters (and what are objects being clustered at first hand), and how an overlapping model would help.
Also, "more distinct accessory REGIONS" rather than "genomes" would be more appropriate, otherwise meaning of “conserved core genome” needs to be specified.
Also, "core genome(s)" and "single contig(s)" should be in the same (singular or plural) form.

6. On line 232, notation “|x|” for the total genome length is weird, since x denotes a specific position.

7. On line 240, “trans” and “cis” read-pairs are introduced relatively to set of contigs, while these notations have already been used before in a different context.

8. Have the authors tried to work with real E.coli strains? Many finished reference genomes are available at various evolutionary distances.

9. The “1/e” community abundance profile is not explained.

10. On line 402, should it be “r=0.61” ?

11. On lines 404-407, increase in edge complexity is stated as a bad feature, which is misleading, since edges give the information for the clustering! At the same time, while it is stated that “increased 3C sampling depth can significantly improve the quality of clustering solutions”, it is not explained why it should help if the edge complexity stopped increasing?

12. On lines 458-459, alternative “more technically complicated HiC protocol” is mentioned but neither referenced nor explained.

13. What is “graphical order” in “Graphical complexity (HL) exhibits a similar turning point to L50 and for the 3C contig graph is dominated by graphical order”?

14. Addition of “sanity-check” dataset with distant genomes and inter-chromosome read-pair background is highly suggested.


Overall, this is an interesting work, clearly showing the limitations of simple graph clustering techniques to the problem of deconvolving microbial communities. But the final message:
"We recommend that future work focus on the application of recent advances soft-clustering methods"
sounds questionable to me. After reading the paper, I would argue that better deconvolution techniques should be developed, rather than staying limited to the general-purpose graph clustering algorithms! It seems that graph clustering techniques will always give many precision errors. For example, imagine conservative regions and 2 strains with long single strain-specific region in each. They will be very well connected through conservative regions, but two long fragments not connected by linkage info should not be placed into the same cluster at all costs!

·

Basic reporting

In general the introduction is adequate and the methods section is for the most part clear, providing sufficient detail. However, I found the results section to be insufficiently contextualised. While as a reviewer I had time to read the manuscript from start to finish, I do not expect that the average reader has such a luxury. Thus, the results section should briefly explain the experimental design before presenting the results of those experiments and their interpretation.

In general the figures and figure legends could be more clearly presented. For instance in Figure 5 the labels are often unreadable, and it is not immediately clear what “individuals” and “variables” refer to. In particular:

* Figure 3: The acronym “BL” should be spelled out.
* Figure 4 y-axis labels should have units (where applicable).
* Figure 4c: The dashed lines should be included in the legend. The dashed lines are suggested in the figure to be first order derivatives of the solid lines if I understand correctly. However, this appears to be incorrect as the slope of the solid lines is clearly positive around 97% ANIb where the “first derivative” is graphed as negative.
* Figure 5 & 6. It is unclear what “nhic” refers to. I would guess this is the same as “n3c” but this should be clarified, and acronyms used more sparsely in the figure legends.

The repository of code at https://github.com/koadman/proxigenomics should be mentioned in the main text of the manuscript. I was only able to find reference to it in the Declarations section. The focus of the manuscript is the interpretation of the results of this code as opposed to the code itself, but nonetheless it would be helpful if readers were instructed as to how central parts of the study relate to the code base. Also, it would be helpful if the base directory of the repository contained a README which outlined the broad structure of the repository as a whole.

Further minor comments:
* 46: cell lysis is not the only process that happens during the DNA purification step.
* 52: Assembly algorithms do not require this assumption to be uniformly true as this sentence implies.
* 56: Assembly is not generally conceptualised in terms of a “system” in this sentence, making this sentence unclear.
* Paragraph starting 69: This should be better structured, introducing the basic idea of the family of methods before listing the different sub-types.
* 73: “Thus” does not make sense here.
* 84: The implication here is that one read comes from each side of proximity ligation. As in the discussion (line 493), this is not true for all cases. While this simplification could be considered appropriate for the manuscript as a whole, it should be explained in the introduction.
* 90: It is unclear whether the author suggest deconvolution of raw metagenome reads or contigs/scaffolds. Certainly, these techniques cannot deconvolute the actual cells themselves (as I believe the authors do not intend to imply).
* 116 and throughout: the term “contig graph” is not inappropriate, but for the sake of avoiding confusion with the more standard meaning of this term in genome assembly I suggest an alternative name be used.
* 129: “objects allowed”: missing “are”
* 140: “would be” should be replaced by “are”
* 143: It seems self-evident that validation measures not be primarily concerned with algorithmic (time) complexity of the algorithms but instead their utility in providing a solution to the problem at hand. Thus the introductory sentences are not needed.
* 214: This paragraph is confused and overly long if the main point of the paragraph is simply that 3C reads need to be mapped rather than extracted from the de-Bruijn graph.
* 281: It should be made more explicit that the 3C reads were simulated using the full genomic sequences rather than the contigs. I also suggest that the read simulation is sufficiently important that it should be included in its own section toward the beginning of the Methods rather than in the catch-all “Pipeline Design” subsection.
* 362: The HL graph complexity measure is not given a name (beyond “HL” and “non-parametric entropy”) and I could not find HL (by that name) discussed in either of the two references given. It may be helpful if the original article proposing the measure be cited as opposed to articles which discuss them.
* 374 and elsewhere in the Results: when discussing the results of experiments, it should be done in past tense rather than present tense.
* 414: p-values should be reported as e.g. “<1x10^92” rather than “2.77x10^-93” as the exact values are not especially relevant.
* 426: The first paragraphs of the discussion appear to be more like results than discussion.
* 545: “increase” should be plural.

Experimental design

The approaches presented in the article appear sound. My only concern is that only the uniformly distributed community structure was used to assess the clustering algorithms. Uniformly distributions of microbial community abundances are very rare in nature, making interpretation of these data difficult if these clustering methods are to be used in the context of 3C data derived from real microbial communities.

Validity of the findings

The largest concern I have with the approach is that no error was incorporated into the simulation of 3C pairs. Not including erroneous links between contigs risks making the simulation overly simplistic. While I am unaware of any simulator for metagenomic 3C data, analysis of clustering algorithms without error may hide complexities that hinder algorithm performance on real data. For instance, while not directly affecting any of the clustering algorithms, the graph complexity measure would be affected by the presence of spurious edges in the adjacency matrix. If as the authors claim in the discussion, “a simple low frequency threshold removal … could in principle solve the problem”, then this approach could be implemented to alleviate these concerns.

The discussion and conclusion sections tend to imply that the 3C clustering/binning problem can be conceptualised as a graph-based problem, but reducing the data to fit this model is not without information loss. For instance, mismatches between the sequences of the 3C read pairs and the contig to which they map might be used to infer that the contig is of mixed origin. This information is destroyed by converting the data into a collection of nodes and weighted edges. As I imagine the authors are aware, there are a number of other information sources that do not neatly fit into a graph-based framework. This does not mean that graph-based methodologies developed in the article are without value, but that they should be better contextualised in the discussion.

---

## Round 0.2 · Minor Revisions

Both reviewers have a relatively minor list of 'fixes' for you to work on. Please note especially Ben's suggestion for a 'Limitations and Future Work' section. I agree that this would be helpful. I judge these to be relatively minor revisions and will not send your paper out for further review, but will evaluate the revised draft and letter addressing the concerns myself when it is ready.

Reviewer 1 ·

Basic reporting

The authors improved the paper significantly, it is much readable now and almost ready for publication.

Experimental design

No Comments

Validity of the findings

No Comments

Additional comments

Some minor comments:

(1) page 4, line 153: It is still a tautology (since the cardinality of any set is >=0).
Perhaps, it should be ">" or "\neq" instead of ">=".

(2) page 4, line 159: Mentioning of "1-to-1" is still confusing. It is better to rephrase like "placing a contig into a unique source-genome bin (cluster) is not possible".

(3) page 4, line 181. It makes sense to define the "harmonic mean".

(4) page 6, lines 267-268. The meaning of the formula is unclear. Perhaps, it should be something like "(n_i,n_j) is an edge \iff i \neq j".

·

Basic reporting

I feel that the manuscript would benefit from some further drafting. Some specific comments:

Fig 4: The HL label in the figure is mangled.
Figure 5: The points are stated as “3C-contigs”. I am unclear exactly what is represented in the figure but I suppose the authors mean contig-graphs? If so, the principle components are being derived from a matrix of contig graph vs (sweep parameters from Table 1 where the algorithm choice column is not a factor but given the numeric value of its Fb3)?
Fig 5: It is unclear why some sizes are different
Fig 5: It should be stated that the percentages after the axis labels show the percent of variation explained, if that is what they show.
Figure 6 caption: misspelling of” increased”.
Figure 6. The caption only refers to Fb3, when it should also refer to Pb3 and Rb3.
143: The first sentence still feels out of place, and the paragraph would flow better if it was moved to after “multiple aspects must be considered”.
442: There is no d panel in Figure 4.
479: Use of the word “parameters” suggests that the runtime is an input of the sweep, it should be replaced/rephrased.
487: The sampling depth is unitless, when it shouldn’t be. It might be more directly informative if the number of reads per genome length was reported.
606: It might be clearer to refer to “contig length-weighted” rather than “object-weighted”.
609: I think it can reasonably be concluded that the same conclusions hold for Archaea.

Experimental design

No comment.

Validity of the findings

Some further (but brief) enumeration of the assumptions used in the simulation in the “Limitations and Future Work” section would be helpful e.g. the simplicity of the community, error free reads, etc.

In regards to the software itself, nestly is missing from the list of dependencies (though it is in the pip install list), and I suspect that perl is also needed. Also, the install instructions suggest that “community” is needed, but this python module seems to be an unrelated module, perhaps another was intended. Either way, I was unable to test because no sample.xml file was given and I was unsure of its structure. It would also be helpful to indicate that the “-j N” refers to the number of cores used.

---

## Round 0.3 · accepted · Accept

Thanks again for your careful consideration of the reviewers' comments. I think the manuscript is ready to go at this point.